# Active School Breaks and Students’ Attention: A Systematic Review with Meta-Analysis

**DOI:** 10.3390/brainsci11060675

**Published:** 2021-05-21

**Authors:** Álvaro Infantes-Paniagua, Ana Filipa Silva, Rodrigo Ramirez-Campillo, Hugo Sarmento, Francisco Tomás González-Fernández, Sixto González-Víllora, Filipe Manuel Clemente

**Affiliations:** 1Department of Physical Education, Arts Education, and Music, Faculty of Education of Albacete, University of Castilla-La Mancha, 02071 Albacete, Spain; 2N2i, Polytechnic Institute of Maia, 4475-690 Maia, Portugal; anafilsilva@gmail.com; 3The Research Centre in Sports Sciences, Health Sciences and Human Development (CIDESD), 5001-801 Vila Real, Portugal; 4Department of Physical Activity Sciences, Universidad de Los Lagos, 8320000 Santiago, Chile; r.ramirez@ulagos.cl; 5Centro de Investigación en Fisiología del Ejercicio, Facultad de Ciencias, Universidad Mayor, 7500000 Santiago, Chile; 6University of Coimbra, 3004-531 Coimbra, Portugal; hg.sarmento@gmail.com; 7Research Unit for Sport and Physical Activity, Faculty of Sport Sciences and Physical Education, 3004-531 Coimbra, Portugal; 8Centro de Estudios Superiores Alberta Giménez, Department of Physical Activity and Sport Sciences, Pontifical University of Comillas, 07013 Palma, Spain; francis.gonzalez.fernandez@gmail.com; 9Department of Physical Education, Arts Education, and Music, Faculty of Education of Cuenca, University of Castilla-La Mancha, 17071 Cuenca, Spain; sixto.gonzalez@uclm.es; 10Escola Superior Desporto e Lazer, Instituto Politécnico de Viana do Castelo, Rua Escola Industrial e Comercial de Nun’Álvares, 4900-347 Viana do Castelo, Portugal; Filipe.clemente5@gmail.com; 11Instituto de Telecomunicações, Delegação da Covilhã, 1049-001 Lisboa, Portugal

**Keywords:** physical activity, exercise, attention, attentional bias, arousal, randomized controlled trials, non-randomized controlled trials, cross-over studies, systematic review, meta-analysis

## Abstract

School physical activity breaks are currently being proposed as a way to improve students’ learning. However, there is no clear evidence of the effects of active school breaks on academic-related cognitive outcomes. The present systematic review with meta-analysis scrutinized and synthesized the literature related to the effects of active breaks on students’ attention. On January 12th, 2021, PubMed, PsycINFO, Scopus, SPORTDiscus, and Web of Science were searched for published interventions with counterbalanced cross-over or parallel-groups designs with a control group, including school-based active breaks, objective attentional outcomes, and healthy students of any age. Studies’ results were qualitatively synthesized, and meta-analyses were performed if at least three study groups provided pre-post data for the same measure. Results showed some positive acute and chronic effects of active breaks on attentional outcomes (i.e., accuracy, concentration, inhibition, and sustained attention), especially on selective attention. However, most of the results were not significant. The small number of included studies and their heterogeneous design are the primary limitations of the present study. Although the results do not clearly point out the positive effects of active breaks, they do not compromise students’ attention. The key roles of intensity and the leader of the active break are discussed. INPLASY registration number: 202110054.

## 1. Introduction

Active breaks (ABs) are currently gaining attention within the educational context [1]. ABs consist of short periods (usually between five and 15 min) of classroom-based physical activity (PA) [2], which are integrated into the routine of the class [1,3]. These can be implemented by the teacher [2] during or between academic instructions [4]. Compared to other kinds of school-based PA interventions, ABs show some advantages. For example, (i) they do not require special spaces or equipment, (ii) teachers can choose when to utilize ABs according to their lessons’ necessities [2], and (iii) they are not too time-consuming for practical use [5].

Some authors consider ABs to be an effective approach to promote PA with the final aim of improving students’ health since school time represents an ideal setting for such purposes [6]. There is evidence confirming that school-based PA interventions increase students’ PA levels [7,8]. In fact, the scientific literature suggests that the brain learns better when active methodologies (active role of students) are implemented instead of passive methodologies or traditional lessons [9]. This is of special interest nowadays because most young students do not meet the PA guidelines recommended by the World Health Organization [10]. Additionally, there is extensive research showing that PA interventions can improve students’ cognitive, metacognitive, and academic outcomes, such as working memory, attention, processing speed, and academic performance [11,12]. Both the acute and chronic effects of school-based PA interventions on cognitive and academic performance have been extensively reviewed [1,3,12,13]. Although their positive effects are not completely clear for all those variables (e.g., attention, processing speed, or academic performance), it seems that increasing the amount of school time spent on PA does not compromise students’ cognitive or academic performance. Therefore, school-based PA interventions are promising practices when appropriately implemented [14].

Among the cognitive outcomes addressed, attention is of great relevance for students since it plays a key role in learning [15] and academic achievement [16]. Conceptualizing attention is not easy due to the myriad of concepts that it involves. Therefore, in the present study, following Janssen et al. (2014) [17], we did not focus on a single measure of attention but instead considered attentional outcomes objectively measured within AB research.

Some authors perceive attention as a process of exerting mental effort on specific stimuli [18], while for others, it is like a “gate” that manages the input of information into conscious awareness [6]. Notwithstanding, most researchers agree with the multi-component nature of attention [19]. This is reflected in the numerous different tasks used to measure attention in previous research [17], such as concentration tasks, time-on-task behaviors, or even electroencephalography.

It has been hypothesized that the effects of PA on attention, both from acute and chronic points of view, have a physiological basis (e.g., cardiovascular hypothesis, intensity of PA, or increases in cerebral blood flow and the number of neurotransmitters) [20]. There is sufficient evidence to suggest that ABs improve cognition, especially attention [20,21,22]. Furthermore, previous evidence has shown that ABs can improve students’ attention [7,8,23]. However, results are still heterogeneous [2] and require further confirmation [3].

The inconsistent results presented in the literature could perhaps be explained by differences in the factors considered from a study design, how attention is measured, or the inclusion of samples representing different age groups [6] or cardiovascular fitness levels [6], just as happens in research on overall cognitive performance [2,12]. Moreover, in one study [24], it was suggested that different results on cognitive effects arise from differences in AB characteristics (e.g., cognitive engagement or complexity, intensity, duration).

Regarding the acute effects of AB duration, a recent meta-regression analysis suggested that shorter PA bouts may be more effective than longer ones for improving attention [12]. However, other research has indicated that longer bouts (i.e., >20 min) showed greater effects [20]. Since other studies did not find differences regarding the duration of PA bout [25], this topic requires further research to investigate the optimal duration of ABs.

Researchers have already highlighted the importance of investigating the duration of the cognitive benefits that remain after a PA bout, which is difficult to establish since post-test timings vary widely across studies, and most studies do not correctly report this information [1].

Finally, the person responsible for delivering the AB might also influence the characteristics of PA, especially regarding the intensity and student engagement [26,27]. In their review, Daly-Smith et al. (2019) [1] reported that the highest proportion of time spent on moderate-vigorous physical activity (MVPA) in active lessons was associated with the researcher-led intervention. Similarly, Watson et al. (2017) [3] pointed out that programs presented a higher fidelity to the required intensity when research staff was responsible for the intervention. These studies highlight the importance of the intervention deliverers’ qualifications, which has not yet been clarified in the AB literature.

For more than a decade, students’ attention deficits have been a significant concern of teachers [28]. ABs seem to be a promising way to enhance students’ attentional levels in the class. However, several questions remain to be answered. To the best of our knowledge, there is only one previous meta-analysis that examined this issue [2], and it focused only on overall cognitive- or academic-related outcomes in 6- to 9-year-old students.

Therefore, the aim of this systematic review with meta-analysis was to scrutinize and synthesize the literature related to the effects of ABs (compared to control conditions) on the attention of students (of any age). We also addressed some possible moderators that previous research pointed out as relevant to the effects of ABs on cognition.

## 2. Materials and Methods

This systematic review (with meta-analysis) followed established international guidelines [29,30]. The protocol was published in INPLASY (International Platform of Registered Systematic Review and Meta-analysis Protocols) with the identification number of 202,110,054 and DOI 10.37766/inplasy2021.1.0054.

### 2.1. Eligibility Criteria

According to the Participants, Intervention, Comparators, Outcomes, and Study design (P.I.C.O.S.) approach, the inclusion and exclusion criteria for this systematic review and meta-analysis can be found in Table 1.

### 2.2. Information Sources

Five electronic databases (PubMed, PsycINFO, Scopus, SPORTDiscus, and Web of Science) were searched for relevant publications prior to 12 January 2021. Keywords and synonyms were entered in various combinations: (“activ* break*” OR “physical break*” OR “physical activity break*” OR “exercise break*” OR “brain break” OR “brain hacking” OR “movement learning” OR “active learning”) AND (student* OR class* OR school*) AND attent*. Additionally, the reference lists of included studies were manually searched to identify potentially eligible studies not captured by the electronic searches. All records were screened by two researchers (AIP and FTGF).

### 2.3. Data Extraction

A data extraction was prepared in Microsoft Excel sheet (Microsoft Corporation, Readmon, WA, USA), similar to the Cochrane Consumers and Communication Review Group’s data extraction template (Group, 2016). The Excel sheet was used to assess inclusion and exclusion requirements and subsequently tested for all selected studies. The process was independently conducted by two authors (AIP and HS). Any disagreement regarding study eligibility was resolved in a discussion with a third author (FTGF) when necessary. Full text articles excluded and the reasons for doing so were recorded (see Table A1 in Appendix A). All the records were stored in the sheet.

### 2.4. Data Items

The following categories of information were extracted from included articles: (i) randomization unit, design, number of participants (n), age group (schoolchildren, young adults or both), sex (men, women or both); (ii) fitness of participants; (iii) identification of ABs (time, duration, weekly and/or daily frequency, intensity and type of PA, academic content, the person who is responsible for the AB, and protocol), (iv) treatment fidelity, (v) measurement of attention (i.e., task), (vi) time of measurements (pre and post) and (vii) effect measured (i.e., acute effects vs. chronic effects).

### 2.5. Assessment of Methodological Quality

The methodological quality of studies was assessed using the *Revised Cochrane risk-of-bias tool for randomized trials* (RoB 2) for randomized controlled trials (RCTs) [31], as well as the supplements for cluster randomized trials (CRTs) [32] and for cross-over trials [33]. For non-RCT, the *Cochrane* risk of bias tool for non-randomized studies of interventions (ROBINS-I) scale was used [34]. These tools include a minimum of 21 items that enable the assessment of the risk of bias (i.e., “low risk”, “some concerns”, or “high risk”) of several dimensions that vary according to the study design (namely, bias arising from the randomization process, bias due to deviations from intended interventions, bias due to missing outcome data, bias in measurement of the outcome, and bias in selection of the reported result). An ‘intention-to-treat’ effect approach was followed for all the assessments, which implies that the interest focused on the effect of assignment to the interventions. This approach was followed because there was a wide variety of study designs and protocols could not be reviewed in most of the cases. Altogether an overall level of risk of bias per study was computed. Risk of bias assessments were based on the published articles, which were accompanied with the trial protocols in two studies [6,35]. Two of the authors (AIP and HS) independently screened and assessed the included articles. Discrepancies were solved by consensus between the two authors without the need for assistance from a third author.

### 2.6. Statistical Analyses

Meta-analyses were performed if at least three study groups provided pre-post AB-related data for the same measure. Using a random-effects model, the means and standard deviations (SD) for dependent variables were used to calculate effect sizes (ES; Hedges’ g) for each outcome in AB treatments and control conditions. When means and SDs were not available, they were obtained from 95% confidence intervals (CIs) or standard error of mean (SEM), using Cochrane recommended formulas. Data were standardized using post-intervention SD values. The ES values are presented with 95% confidence intervals (CI). Calculated ES were interpreted using the following scale: <0.2, trivial; 0.2–0.6, small; >0.6–1.2, moderate; >1.2–2.0, large; >2.0–4.0, very large; >4.0, extremely large [36]. Heterogeneity was assessed using the *I^2^* statistic, with values of <25%, 25–75%, and >75% considered to represent low, moderate, and high levels of heterogeneity, respectively [37]. The risk of bias was explored using the extended Egger’s test [38]. To adjust for publication bias, a sensitivity analysis was conducted using the trim and fill method [39], with L0 as the default estimator for the number of missing studies [40]. All analyses were carried out using the Comprehensive Meta-Analysis software (version 2; Biostat, Englewood, NJ, USA). Statistical significance was set at *p* ≤ 0.05.

## 3. Results

### 3.1. Study Identification and Selection

The database search retrieved 1809 titles, which were exported to reference manager software (EndNoteTM X9, Clarivate Analytics, Philadelphia, PA, USA). Duplicates (520 references) were subsequently removed either automatically or manually. The remaining 1289 articles were screened for their relevance based on titles and abstracts, resulting in the removal of a further 1244 studies. Following the screening procedure, 45 articles were selected for in-depth reading and analysis. After reading full texts, a further 36 studies were excluded due to not meeting the eligibility criteria (Table A1). Finally, nine studies were selected for the further analysis together with another seven studies that were identified from other sources, reaching a total of 16 included studies (Figure 1), involving 3383 participants between 6 and 13 years old. Due to the limited number of studies included into the review for each attentional outcome (e.g., global attention, selective attention, inhibition, etc.), results from participants of all ages included were grouped together despite this age range involves different stages of development.

### 3.2. Study Characteristics

Eleven studies followed a parallel-groups design with seven RCTs [19,35,41,42,43,44,45], three CRTs [24,46,47], and one non-randomized [48] (Table 2). Additionally, the remaining five studies followed a cross-over design (three single-group design studies [6,23,49] and two CRTs [50,51]) (Table 3).

The protocols’ characteristics for all studies can be found in Table 4. Most of the studies addressed the acute effects of ABs [6,19,23,35,41,42,43,44,45,49,50,51], while only four addressed chronic effects [24,46,47,48]. Considering both types of effects, AB duration varied from 4 [23] to ≈ 25 min [41]. Interventions addressing chronic effects were applied to two ABs per week over two weeks [24] to five ABs per week over ten weeks [46]. Overall, ABs consisted of aerobic or coordinative moderate PA (MPA) [6,23,35,41,46,50], vigorous PA (VPA) [6,44,45], or MVPA [19,41,42,43,47,49]. Two studies reported that the registered intensity was lower (i.e., light or light-to-moderate) than expected [24,51], and one did not report intensity-related data [48]. Regarding the type of PA, nine studies included cognitively engaging PA conditions (i.e., combined activities, games, dancing, or coordinative exercises) [19,24,41,42,43,47,48,49,51], with two of them relating the PA to academic contents [24,49]; the other interventions included aerobic PA. Most of the interventions were delivered by the researchers [6,19,23,24,35,41,42,43,45,51]. Five of them were by the classroom teachers [6,47,48,49,50], and four of them also relied on videos to guide the ABs [24,35,48,51].

### 3.3. Methodological Quality

The overall methodological quality of the intervention studies can be found in Table A2. Nine studies were assessed as having some concerns in their overall RoB 2 quality scale, and eight were assessed as high risk of bias. The score for the only study assessed by ROBINS-I was critical [48]. None of the studies achieved low risk of bias. Methodological assessment revealed issues on the quality of the bias in the information reported on the randomization process, the reporting of possible deviations from the intended interventions and the selection of the reported result.

### 3.4. Active Breaks: Effects on Attention

Due to the multi-component nature of attention, a wide variety of attention-related outcomes were reported among included studies (e.g., global attention; selective attention). Table A3 shows a synthesis of the outcomes according to the task and their scoring. Overall, results from the 11 parallel groups design studies (Table 2) showed that the effects of ABs on attention were mainly positive or non-significant; no negative effect of ABs was found. In addition, results from the five cross-over design studies (Table 3) showed similar results, with positive or no effects on attentional outcomes and without negative results. The results for each outcome are synthesized in the following sections.

#### 3.4.1. Effects on Accuracy

Accuracy was only measured with the d2 test. Acute positive effects were found only in an aerobic MPA AB intervention on 9 to 11-year-old students [23]. No other acute [19,45] nor chronic effects were found [48]. Meta-analyses could not be run since there were less than three studies per analysis.

#### 3.4.2. Effects on Inhibition

Inhibition was measured in four studies through the flanker task, the Stroop task, and the ANT tests. Only one study found acute favorable effects after a 20-min cognitively engaged AB intervention [42]. The remaining studies on acute [41,43] or chronic [47] effects found no significant results.

Regarding the meta-analysis, four studies provided data for inhibition (i.e., reaction time results from flanker and ANT) involving six experimental and four control groups (pooled *n* = 900). There was a trivial effect of AB on inhibition (ES = 0.08; 95% CI = −0.07 to 0.23; *p* = 0.293; *I^2^* = 12.0%; Egger’s test *p* = 0.576; relative weight of each group: 9.9 to 44.7%; Figure 2). After study-by-study and group-by-group sensitivity analyses, no significant changes in results were noted (i.e., *p*-value remained at > 0.05, mean ES = 0.02 to 0.14). No significant sub-group differences (*p* = 0.342) were identified between acute (ES = 0.14; 95% CI = −0.07 to 0.35; within-group *I^2^* = 13.8%, five study groups) and chronic effects (ES = 0.01; 95% CI = –0.18 to 0.19; within-group *I^2^* = 0.0%, one study group).

#### 3.4.3. Effects on Concentration

Eight studies reported an index of concentration performance, which has been mainly measured by the d2 and FACES tests. Regarding the acute effects, one of them found positive effects of 12-min VPA on 9 to 10-year-old students’ concentration [44]. On the other hand, positive chronic effects after four and 10 weeks of intervention were found in two of four studies [46,48]; however, the four-week study of these was assessed as high risk at ROBINS-I [48]. No other significant effects were found.

Six studies provided data for concentration, involving nine experimental and six control groups (pooled *n* = 881). There was a trivial effect of AB on concentration (ES = 0.19; 95% CI = −0.08 to 0.46; *p* = 0.161; *I^2^* = 63.5%; Egger’s test *p* = 0.581; relative weight of each group: 8.2 to 17.9%; Figure 3). A sensitivity analysis according to ROBINS-I was conducted, removing the study of Buchele et al. (2018) [48], with no significant changes in results. However, a study-by-study sensitivity analysis, removing the study of Schmidt et al. (2019) [24], revealed a small effect of AB on concentration performance (ES = 0.34; 95% CI = 0.20 to 0.48; *p* < 0.001). No significant sub-group differences (*p* = 0.627) were identified between acute (ES = 0.27; 95% CI = −0.07 to 0.60; within-group *I^2^* = 0.0%, four study groups) and chronic effects (ES = 0.14; 95% CI = −0.29 to 0.56; within-group *I^2^* = 81.2%, five study groups).

#### 3.4.4. Effects on Selective Attention

Six studies measured selective attention with the d2, FACES and the Sky-Search task in TEA-Ch tests. With the exception of one study [23], the other three studies that measured acute effects reported positive results, with ABs varying between 12 and 20 min of MPA or VPA. Of note, one of the interventions only reported favorable results on the group that participated in two ABs during the same morning and no differences between one AB and no AB [35], while other reported only benefits for the AB of MPA and not for the VPA AB [6]. In addition, greater benefits were reported among low-income students [45]. Regarding the chronic effects, results were similar to the concentration’s results.

Four studies provided data for selective attention, involving five experimental and four control groups (pooled *n* = 395). There was a moderate effect of AB on selective attention (ES = 0.61; 95% CI = 0.41 to 0.82; *p* < 0.001; *I^2^* = 0.0%; Egger’s test *p* = 0.036 (corrected values: ES = 0.67, 95%CI 0.45 to 0.88); relative weight of each group: 7.2 to 41.1%; Figure 4). A sensitivity analysis according to ROBINS-I was conducted, removing the study of Buchele et al. (2018) [48], with no significant changes in results (i.e., *p*-value remained at <0.001, mean ES = 0.59). Similarly, after study-by-study and group-by-group sensitivity analyses, no significant changes in results were noted (i.e., *p*-value remained at <0.001, mean ES = 0.48 to 0.65). No significant sub-group differences (*p* = 0.963) were identified between acute (ES = 0.55; 95% CI = 0.10 to 1.00; within-group *I^2^* = 43.5%, three study groups) and chronic effects (ES = 0.54; 95% CI = 0.23 to 0.84; within-group *I^2^* = 0.0%, two study groups).

#### 3.4.5. Effects on Shifting

Shifting was only assessed in three studies [41,42,43] that used the flanker task to test acute effects. None of them found any acute effect after ABs. In the meta-analyses, the three studies provided data, involving five experimental and three control groups (pooled *n* = 441). There was a trivial effect of AB on shifting (ES = −0.18; 95% CI = −0.52 to 0.15; *p* = 0.286; *I^2^* = 65.7%; Egger’s test *p* = 0.229; relative weight of each group: 19.1 to 21.8%; Figure 5). After study-by-study and group-by-group sensitivity analyses, no significant changes in results were noted (i.e., *p*-value remained at >0.05, mean ES = −0.29 to 0.27).

#### 3.4.6. Effects on Sustained Attention/Vigilance

ANT, d2 and PVT tests were employed in each of the three studies that measured sustained attention/vigilance [47,48,49]. No acute effects were found [49]. On the other hand, positive chronic effects after a four-week intervention were found [48], but it presented a high risk at ROBINS-I. Meta-analyses could not be run since there were less than three studies per analysis.

#### 3.4.7. Effects on Other Outcomes

Only one study [47] measured orienting, in 9 to 12-year-old students, before and after a 9-week intervention of daily MVPA cognitive ABs. No chronic effects were found. In addition, one study, of 8 to 11-year-old students, reported a global outcome of attention by a compendium of different executive function tasks [50]. This study found favorable acute effects after 10-15 min of AB involving aerobic MPA. Due to the reduced number of studies reporting data for these outcomes, a meta-analysis was precluded.

## 4. Discussion

### 4.1. Discussion of Evidence

The present systematic review with meta-analysis scrutinized and synthesized the literature related to the effects of ABs on students’ (of any age) attention when compared to control conditions. The results do not point to any clear acute or chronic effects of ABs on students’ overall attention, although some positive effects were found in terms of accuracy, concentration, inhibition, sustained attention, and (especially) selective attention. The meta-analysis revealed no statistical differences between AB and control groups regarding inhibition (Figure 2), concentration (Figure 3), or shifting (Figure 5). The trends for inhibition and concentration favored AB groups, and the trend for shifting favored the control groups. Nevertheless, all three meta-analyses included zero in their confidence intervals; therefore, no solid conclusions can be drawn. However, for selective attention, there was a significant difference between the AB and control groups (Figure 4) in favor of the former. These results will be discussed. In addition, overall, ABs did not compromise students’ attention.

As a first approximation to the problem, we suggest that the small number of positive effects [6,23,35,42,44,45,46,48,50] in the different included outcomes could be attributed to the fact that performing any type of exercise provokes neurophysiological changes in the brain [52]. Nevertheless, there is much heterogeneity and a wide variety of ABs protocols encountered (i.e., durations ranging from 4 to 20 min; intensities of exercise ranging from moderate and vigorous; the inclusion of various types of PA such as aerobic, anaerobic, and muscular resistance; and the use of specific cognitive tasks to assess attention such as d2, ANT, and the flanker task). These differences do not provide clear evidence and have sparked controversy due to the non-existence of general guidelines for applying and implementing ABs.

Regarding the acute exercise paradigm, positive effects were observed only for accuracy [23], concentration [44], inhibition [42], and selective attention [6,35,45]. This suggests that cognitive activities performed after exercise lasting 4 to 20 min could produce overall benefits to students’ attention. In addition, regarding the studies including chronic AB interventions, positive effects were observed after 10 weeks in terms of concentration and selective attention [46]. Likewise, positive chronic effects on sustained attention were found after a four-week intervention [48].

Despite the lack of support from the meta-analyses, the positive findings found regarding selective attention are in line with the study of Donnelly et al. (2016) [53], who showed that routinely practicing PA in schools enhances cognitive performance. In fact, chronic exercise positively influences different attention processes in children [54]. In all cases, an argument could be made for the importance of studying additional moderators since these could influence the effects of exercise [20]. However, the small number of studies found per outcome did not allow us to make robust distinctions about the effects according to the moderators [20].

Despite the lack of clear moderators explaining the relationship between exercise and cognitive function, the results suggest that the intensity of the exercise used in ABs plays a fundamental role in the literature exploring the specific effects of PA on cognition. In fact, the current research suggests that some attentional outcomes improved after ABs at MVPA intensities (40 to 80 of VO_2_max). However, many studies did not monitor the intensities of ABs, nor did they measure the magnitudes of the changes in some physiological mechanisms (e.g., brain-derived neurotrophic factor, catecholamines, increased cerebral blood flow) [20,55] to predict their possible effects on behavior and cognitive performance.

Nevertheless, in most studies, ABs were carried out in the classroom and never under laboratory conditions. For this reason, measuring exercise intensities with a large sample is a truly complex matter. In fact, objective instruments (either a heart rate monitor [19,35,41,42,43,45,51] or accelerometry [6,24,47,49]) were used to calculate the loads of ABs. However, researchers have also relied on subjective measures controlled by the teachers—in some cases, the measures were simply not registered [23,44,46,48,50]. In all cases, a potential and valid proposal might be the use of the subjective perception of effort (e.g., Egger et al. 2018 [41], Schmidt et al. 2016 [19]). This approach, which helps calculate metabolic changes during exercise, could be an effective option to use in school children [56,57].

In light of the above discussion, another key factor that might moderate the effects of ABs is the person who applies the AB. In this sense, it would be appropriate for physical education teachers to be responsible for applying ABs in all interventions [1,27]. On the one hand, they have the capacity to guide research proposals since they have a deeper knowledge of training principles involved in any kind of PA. On the other hand, they could provide students and other teachers with techniques for controlling the intensity of the AB in each intervention, which is suggested as being a determinant of outcomes in the present work [26].

### 4.2. Study Limitations

The first limitation of this systematic review is the small number of studies found per outcome and associated effect of PA (i.e., acute or chronic) and the heterogeneity among these studies’ designs. This leads to the second main limitation, which is that the ESs for each effect type cannot rely on a minimum of three studies in all cases. In addition, the heterogeneity was considerable for chronic effects in concentration and acute effects in shifting. Altogether, these limitations indicate that the results should be interpreted with caution.

### 4.3. Practical Implications

The outcomes of this study present implications for incorporating ABs into school lessons to improve students’ attention. Through ABs students can reach higher levels of PA, which promotes a healthy lifestyle.

However, teachers are not usually adequately prepared to carry out ABs throughout the day during class. Thus, teachers should be trained on the correct implementation of ABs and the integration of movement on class days to ensure that ABs positively affect students’ health and cognition. To achieve this, ABs should control physiological measures to objectively calculate exercise intensity. As a result, students might obtain more benefits from ABs if teachers are also trained in the use and interpretation of measures of PA intensity feasible for in-class use, such as the Borg scale [58,59,60].

Finally, as it relates to practical implications, the duration of ABs varied from 4 min to more than 20 min. However, previous research has shown that only exercise of more than 20 min had positive results on cognitive performance [20]. From an educational point of view, it could be thought that adding a 20-min break into current school timetables may compromise the learning time. Therefore, ABs of such a length may not be practical. Additionally, evidence on this matter is not clear, as a recent review found that the duration of PA was inversely related to attentional performance [12].

Notwithstanding, from an academic performance perspective, increasing the amount of school-based PA does not compromise academic achievement and can improve classroom behavior and academic achievement [14]. In addition, as seen in some of the included studies, ABs can include academic content [24,49] and, therefore, could also be included in the learning time. Although more conclusive evidence is needed on this topic, ABs could be included and adapted to different educational contexts and would be effective for improving students’ health and cognitive outcomes as long as the required intensity and time are met.

## 5. Conclusions

There are no clear positive effects of ABs on students’ attention. The heterogeneity in the designs and measurements of the studies and the small number of studies carried out in school environments are the main reasons for the lack of conclusive results. Notwithstanding, it seems that including PA in school time through ABs does not compromise students’ attention, and it could positively affect selective attention.

The intensity and duration of the PA seem to play a key role in cognitive effects. Therefore, efforts should be made to help teachers understand how to motivate their students to reach the correct intensity levels when carrying out an AB.

Even though research on ABs started around a decade ago, clear evidence is still lacking regarding their effects on attention. The results presented here highlight that this topic is still of significant relevance.

## Figures and Tables

**Figure 1 brainsci-11-00675-f001:**
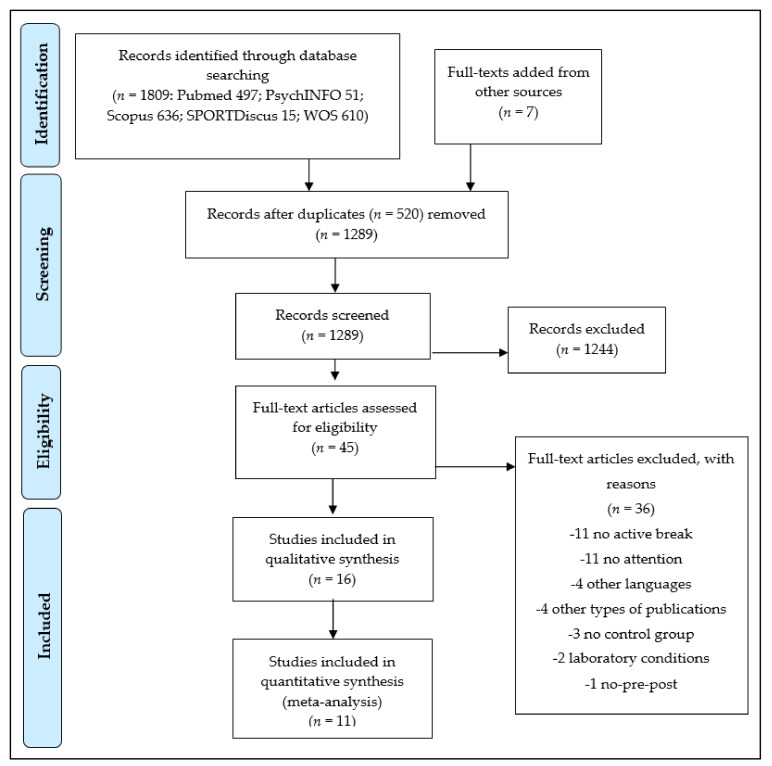
Preferred Reporting Items for Systematic Reviews and Meta-Analyses (PRISMA) flow diagram highlighting the selection process for the studies included in the systematic review.

**Figure 2 brainsci-11-00675-f002:**
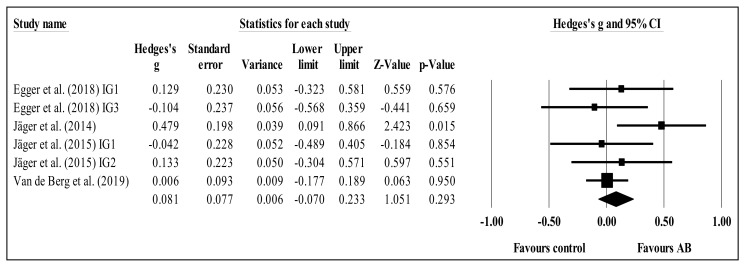
Forest plot of changes in inhibition in school-age students participating in active breaks (AB) compared to controls. Values shown are effect sizes (Hedges’ g) with 95% confidence intervals (CI). The size of the plotted squares reflects the statistical weight of each study. The black diamond reflects the overall result. This trend is not statistically significant. IG: intervention group.

**Figure 3 brainsci-11-00675-f003:**
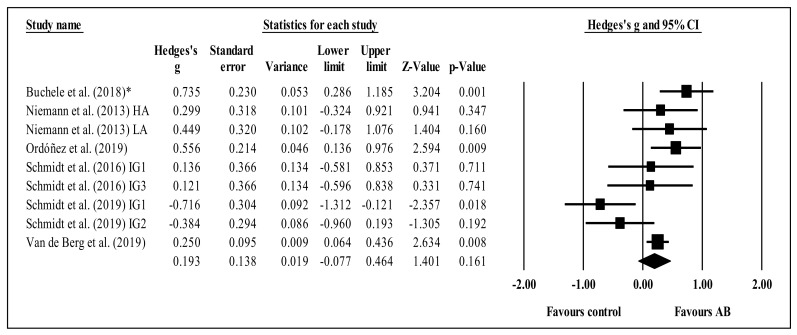
Forest plot of changes in concentration in school-age students participating in active breaks (AB) compared to controls. Values shown are effect sizes (Hedges’ g) with 95% confidence intervals (CI). The size of the plotted squares reflects the statistical weight of each study. The black diamond reflects the overall result. This trend is not statistically significant. IG: intervention group; HA: high-active subgroup; LA: low-active subgroup. * Critical risk in ROBINS-I.

**Figure 4 brainsci-11-00675-f004:**
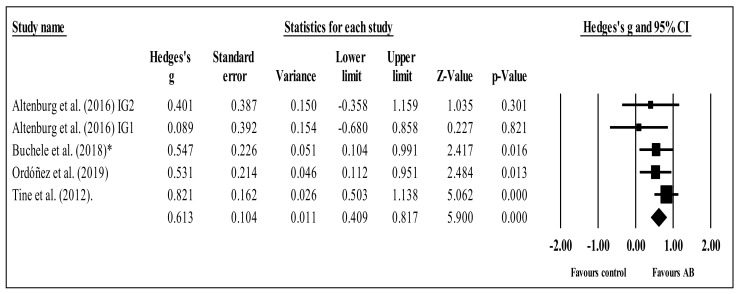
Forest plot of changes in selective attention in school-age students participating in active breaks (AB) compared to controls. Values shown are effect sizes (Hedges’ g) with 95% confidence intervals (CI). The size of the plotted squares reflects the statistical weight of each study. The black diamond reflects the overall result. This is a statistically significant result. IG: intervention group. * Critical risk in ROBINS-I.

**Figure 5 brainsci-11-00675-f005:**
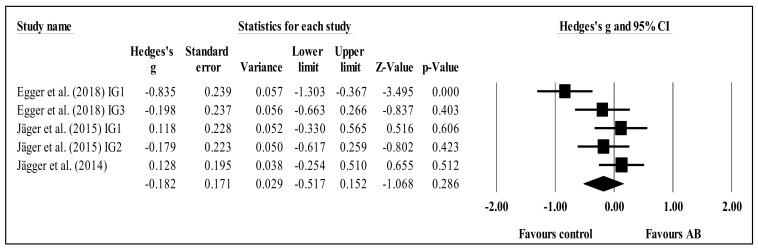
Forest plot of changes in shifting in school-age students participating in active breaks (AB) compared to controls. Values shown are effect sizes (Hedges’s g) with 95% confidence intervals (CI). The size of the plotted squares reflects the statistical weight of each study. The black diamond reflects the overall result. This trend is not statistically significant. IG: intervention group.

**Table 1 brainsci-11-00675-t001:** Inclusion and exclusion criteria following the P.I.C.O.S. approach.

PICOS	Inclusion Criteria	Exclusion Criteria
Population	Healthy students of any age and of any sex from elementary to college educational levels.	Populations other than students (e.g., workers, athletes). Students with a diagnosed mental disease.
Intervention	ABs consisting of short bouts of exercise in class during or between academic lessons (e.g., structured exercises, free exercise).	No ABs (e.g., physical education classes; playing with instruments without allowing PA).
Comparator	Control conditions (passive or non-active breaks with limited PA).	Other forms of physical activity interventions (e.g., physical education lessons).
Outcome	Attentional outcomes (e.g., focused or selective attention, vigilance, inhibitory control) measured before (pre-) and after (post-) ABs or a chronic intervention of ABs.	Outcomes other than attention. No pre-post comparison. Inaccessible pre- or post-intervention data.
Study design	Counterbalanced cross-over design and parallel-groups design.	Study designs that do not allow within-subjects comparisons for both control and AB conditions.
Additional criteria	Original and full-text studies written in English.	Non-original articles (e.g., reviews, letters to editors, trial registrations, proposals for protocols, editorials, book chapters, conference abstracts).

**Table 2 brainsci-11-00675-t002:** Characteristics of the selected studies with a parallel-groups design.

Study Risk of Bias	Design and AB Type	Age (y.o.) Mean ± SD (Range) Academic Level	Sample Size (*n*) and Sex	Attentional Outcomes (Instrument)	Fitness Level	Results
Altenburg et al. (2016) [35] Some concerns	RCT: two IG and one CG. Acute.	NR (10–13) NR	All: 52 (29♀ 33♂)	Selective attention (Sky Search in TEA-Ch)	NR	Children in IG2 (two ABs) had better selective attention than children in IG1 (one AB) or CG. There was no difference between IG1 and CG.
			IG1: 17 (5♀ 12♂)			
			IG2: 20 (9♀ 11♂)			
			CG: 19 (12♀ 7♂)			
Buchele et al. (2018) [48] Critical risk	Non-randomized quasi-experimental: two IGs and one CG. Chronic.	NR (≈10–11) 5th grade	All: 116 (59♀ 57♂)	Accuracy (d2) *	NR	The IG1 increased all attentional outcomes (except accuracy) compared to the CG and concentration and sustained attention compared to IG2 (no AB). There were no differences between IG2 and CG.
			IG1: 31 (14♀ 17♂)	Concentration (d2)		
			IG2: 29 (10♀ 19♂)	Selective attention (d2)		
			CG: 56 (35♀ 21♂)	Sustained attention (d2) *		
Egger et al. (2018) [41] Some concerns	RCT: three IGs and one CG. Acute.	All: 7.94 ± 0.44 (7–9) IG1: 7.99 ± 0.38 IG2: 7.93 ± 0.45 IG3: 7.96 ± 0.50 CG: 7.90 ± 0.44 2nd grade	All: 216 (~106♀ 110♂)	Inhibition reaction time (ms) (flanker task) Shifting reaction time (ms) (flanker task additional block)	Multistage 20m-SRT: IG1: 304.58 ± 123.18. IG2: 284.27 ± 141.16. IG3: 306.43 ± 144.23. CG: 278.55 ± 129.13	A significant, negative effect was found for the CE factor in shifting. No effects were found for the PA factor or the interaction between PA and CE.
			IG1: 59			
			IG2: 53			
			IG3: 50			
			CG: 54			
Jäger et al. (2014) [42] Some concerns	RCT: one IG and one CG. Acute.	7.91 ± 5.05 (months)	All: 104 (57♀ 53♂)	Inhibition reaction time (ms) (flanker task) Shifting reaction time (ms) (flanker task additional block)	Motor fitness: 20m-SRT, 20m sprint test and jump side-to-side.	The AB improved only inhibition, and its effects remained for less than 40 min after the AB. The improvements were suggested to be independent of the participants’ characteristics and stronger among those with higher increases in cortisol.
		(6.83–8.92)	IG: 51 (27♀ 24♂)			
		2nd grade	CG: 53 (30♀ 23♂)			
Jäger et al. (2015) [43] High risk	RCT: three IGs and one CG. Acute.	11.29 ± 6.53 (months) (10.33–12.33) NR	All: 217 (120♀ 97♂) IG1: 54 (35♀ 19♂) IG2: 62 (28♀ 34♂) IG3: 60 (30♀ 30♂) CG: 58 (33♀ 25♂)	Inhibition reaction time (ms) (flanker task) Shifting reaction time (ms) (flanker task additional block)	18-mSRT: VO_2_max (ml/kg/min): Posttest: IG1: 46.77 ± 6.73, IG2: 47.98 ± 6.01, IG3: 46.77 ± 5.96), CG: 47.58 (6.12)	No effects of AB (with and without considering CE) were found. Fitness did not moderate the effects.
Niemann et al. (2013) [44] High risk	RCT: one IG and one CG. Acute.	9.69 ± 0.44 (9–10) IG: 9.65 ± 0.41 CG: 9.74 ± 0.48 4th grade	All: 42 IG: 27 (13♀ 14♂) CG: 15 (7♀ 8♂)	Concentration (d2)	NR	The IG showed better concentration than CG, although both groups improved from pre- to post-test. There was an interaction between group (IG, CG) test (pre, post), and PA level (high, low).
Ordóñez et al. (2019) [46] High risk	CRT: one IG and one CG. Chronic.	11.1 (11-12) 6th grade (Spanish Elementary Education)	All: 89 IG: 45 CG: 44	Concentration (FACES) Selective attention (FACES)	ALPHA. Lower-limb muscle strength (meters): Pretest: IG: 1.36 ± 0.21; CG: 1.38 ± 0.20. Posttest: IG: 1.42 ± 0.21; CG: 1.40 ± 0.21. Coordination (no. jumps): Pretest: IG: 28.33 ± 6.89; CG: 26.40 ± 5.68. Posttest: IG: 30.87 ± 5.68; CG: 27.33 ± 5.90. Cardiorespiratory capacity (min): Pretest: IG: 6.42 ± 0.75; CG:6.46 ± 0.83. Posttest: IG: 5.61 ± 0.68; CG: 6.20 ± 0.75	Significant differences between groups with higher levels of attention in the IG.
Schmidt et al. (2016) [19] Some concerns	RCT: three IGs and one CG. Acute.	11.77 ± 0.41 (11.01–12.98) 5th grade	All: 92 (42♀ 50♂) IG1: 25 (~12♀ 23♂) IG2: 22 (10♀ 12♂) IG3: 25 (11♀ 14♂) CG: 20 (9♀ 11♂)	Accuracy (d2) * Concentration (d2)	NR	No significant effects of ABs or their interactions with CE were found concerning attention. However, high CE interventions had a positive effect on focused attention, and positive affect had a mediational role between CE factor, accuracy, and focused attention, but not for PA.
Schmidt et al. (2019) [24] Some concerns	CRT: two IGs and one CG. Chronic.	9.04 ± 0.70 3rd grade	All: 104 (50♀ 54♂) IG1: 34 IG2: 37 CG: 33	Concentration (d2) (measured after 3rd AB)	NR	Focused attention did not differ between the three groups after controlling for age, step counts, and attention at pretest.
Tine et al. (2012) [45] High risk	RCT: one IG and one CG. Acute	NR (10.33–13.5) 6th–7th grade	All: 164 IG:86 (45♀ 41♂) CG: 78 (40♀ 38♂) (divided by income)	Accuracy (d2) * Selective attention (d2)	NR	The IG improved only regarding selective attention. Moreover, lower-income children exhibited greater improvements than higher-income children.
Van den Berg et al. (2019) [47] For most outcomes: Some concernsFor d2: High risk	CRT: one IG and one CG. Chronic.	IG: 10.8 ± 0.6 CG: 10.9 ± 0.7 (9–12) 5th–6th grade	All: 510 (448 to 467, depending on the outcome). IG: 100 (46♀ 54♂) CG: 100 (47♀ 53♂)	Alerting reaction time (ms) and accuracy (%) (ANT) *Concentration (d2) Inhibition reaction time (ms) and accuracy (%) (ANT ^a^ and Stroop Color Word Task *) Orienting reaction time (ms) and accuracy (%) (ANT) *	18-mSRT: VO_2_max (ml/kg/min): Pretest: IG: 48.1 ± 5.0; CG: 48.0 ± 5.0. Posttest: IG: 48.9 ± 0.2; CG: 48.8 ± 0.2	No intervention effects were detected on any outcome after controlling for pretest score, age, arithmetic performance, class, and school. The IG spent more time in MVPA, but their fitness levels were similar to students in the CG.

AB: active break, ANT: attentional network test; BMI: body mass index, CE: cognitive exertion, CG: control group, CRT: cluster randomized trial, EF: executive functions, IG: intervention group, NR: not reported, PA: physical activity, RCT: randomized controlled trial, SES: socioeconomic status, SRT: shuttle run test; TEA-Ch: Test of Selective Attention in Children. ^a^ Executive control, but it is similar to the other inhibition tasks. Therefore, results were treated as inhibition. ***** Not included in the meta-analysis.

**Table 3 brainsci-11-00675-t003:** Characteristics of the selected studies with a cross-over design *.

Study Risk of Bias	Design and Type of AB	Age (y.o.) Mean ± SD (Range) Academic Level	Sample Size (*n*)/Sex	Outcomes (Instruments/Tasks)	Fitness Level	Results
Hill et al. (2010) [50] Some concerns	CRT counterbalanced with two conditions. Acute.	NR (8-11) 4th–7th grade (Scottish)	All: 1224 (1074 completed three or more of the tests on both weeks)	Global attention: overall performance of different executive functions tests)	-	AB improved attention only among participants who received the intervention in the second period. Improvements were moderated by test and age.
Janssen et al. (2014) [6] High risk	Single group. Three randomized conditions at the group level. Acute.	10.4 ± 0.59 (10–11) 5th grade	All: 123 (61♀ 62♂)	Selective attention (Sky Search in TEA-Ch)	20-mSRT (dichotomized into high or low)	Attention was significantly better in all the conditions than in the ‘no break’ condition. Attention scores were best after the MPA AB. Attention after VPA breaks was better than after no break but was no different than after the passive break. No moderation effect of fitness was detected.
Ma et al. (2015) [23] High risk	Single group (divided). Two randomized conditions at the group level. Acute (mean of several acutes).	NR (9–11) 3–5th grade	All: 88 (44♀ 44♂)	Accuracy (d2) Concentration (d2) Selective attention (d2)	-	Better processing speed scores were reported after no AB. Accuracy improved after the AB. No effects on selective attention were observed following the AB, although accuracy improved.
van den Berg et al. (2016) [51] Some concerns	CRT counterbalanced with two conditions in three different groups. Acute.	11.7 ± 0.7 (10–13) 5th–6th grade	All: 184 (46♀ 54♂) IC1: 66 (47♀ 53♂) IC2: 71 (44♀ 56♂) IC3: 47 (49♀ 51♂)	Concentration (d2)	-	No effects of ABs (LMPA) on attention were reported nor were differential effects of exercise type, after controlling for age and session order. Scores for both conditions improved from day 1 to day 2.
Wilson et al. (2016) [49] Some concerns	Single group. Two randomized conditions at the group level. Acute (mean of several acutes).	11.2 ± 0.6 (≈10–12) 5th–6th grade	All: 58 ♂	Vigilance Mean Reaction Time (ms) and lapses (%) (PVT)	-	There were no significant differences between the AB and no-AB conditions.

AB: active break, BMI: body mass index, CC: control condition, CE: cognitive exertion, CRT: cluster randomized trial, EF: executive functions, IC: intervention condition, NR: not reported, PA: physical activity, PVT: psychomotor vigilance task; RCT: randomized controlled trial, SES: socioeconomic status, SRT: shuttle run test; TEA-Ch: Test of Selective Attention in Children. ***** Not included in the meta-analysis.

**Table 4 brainsci-11-00675-t004:** Protocols of interventions.

Study ID	Type of AB	CG/CC Activity	AB Duration (Min)	Duration and Weekly/Daily Freq.	Time of AB	Intensity and Type of PA	Responsible	Timing of Pre-Test and Post-Test	Fidelity
Altenburg et al. (2016) [35]	**IG1**: One 20-min AB. **IG2**: Two 20-min PA bouts.	**CG**: No PA. Sitting all morning working on simulated school tasks.	20	NA IG1: 1 t-d; IG2: 2 t-d	IG1: after 90 min of sitting. IG2: one AB at the start and another after 90 min of sitting.	MPA. Aerobic. No AC.	Supervising research staff with videos	Pre: At baseline (T0). Post: After 20 min of school, after 130 min; and after 220 min.	HR monitor
Buchele et al. (2018) [48]	**IG1** “Coordinated bilateral PA”. **IG2 *** “Fitbit Only”: Participants wore HR monitors on weekly days with no addition instructions.	**CG**: Usually scheduled school academic instruction periods while wearing plastic wristbands.	6	4 weeks 5 d-w/1 t-d	After 20 min of sedentary behavior.	NA IG1 Coordination.IG2: no PA. No AC.	Teachers with videos	Pre: The previous week the intervention. Post: The week after the intervention.	NR
Egger et al. (2018) [41]	**IG1** “Combo: high CE + high PA”: Running while listening to a song with keywords to perform specific actions and inhibit others. **IG2 *** “Cognition: high CE + low PA”: Sitting while listening and reacting to a song. **IG3** “Aerobic: low CE + high PA”: Running while listening to a song, but without changing the actions performed.	**CG** “Low CE + low PA”: Participants sat comfortably in a circle and listened to an age-appropriate audio-book for 20 min.	≈25 ^a^	NA	Morning (9:25–9:50 a.m.)	MVPA. IG1: Cognitive. IG2: no PA. IG3: Aerobic.No AC.	Researcher	Pre: Before the AB (9:05–9:25 a.m.). Post: Immediately after the AB (9:50–10:10 am).	HR monitors and Borg RPE scale. Perceived CE was also assessed
Hill et al. (2010) [50]	**IC**: Stretching and aerobic PA (e.g., running on the spot, hopping sequences to music).	**CC**: Normal curriculum plan.	10–15	2 weeks 5 d-w/1 t-d	≈30 min after lunch.	MPA. Aerobic. No AC.	Trained teachers	Pre: NR. Post: At the end of the school day.	Teachers’ control
Jäger et al. (2014) [42]	**IG** “EF-specific cognitive engaging PA”: Warm-up with a song, playing tag, and balancing on various objects.	**CG**: 15 min seated on a mat while listening to an age-appropriate story. The last 5 min were spent answering easy questions.	≈20	NA	10:00–10:20 a.m.	MVPA. Cognitive. No AC.	Researcher	Pre: Prior the intervention. Post: Just after the AB. Follow-up 40 min after.	HR monitors
Jäger et al. (2015) [43]	**IG1** “Physical games: PA + CE”: three different cooperative and competitive PA games involving EF. **IG2** “Aerobic exercise”: Short tasks and games with different forms of running. **IG3 *** “Cognitive games: Sedentary + CE”: card game.	**CG**: Sedentary without CE: Participants sat comfortably on a mat and listened to an age-appropriate story.	20	NA	NR	MVPA. IG1: Cognitive IG2: Aerobic. IG3: no PA. No AC.	Researcher	Pre: Just before the intervention. Post: Immediately after the intervention.	HR monitors
Janssen et al. (2014) [6]	**IC1** “MPA-AB”: Walking to and from the PE classroom, jogging, and passing and dribbling a ball. **IC2** “VPA-AB”: Running to and from the PE classroom, running, jumping, and rope skipping.	**CC1**: No break. Participants were not allowed to ask the teacher for help or go to the toilet. **CC2**: Passive break. The teacher read a story to the participants.	15	NA	After an hour of regular cognitive tasks (9:30–10:00 am)	IC1: MPA: Aerobic. IC2: VPA: Aerobic. No AC.	Two researchers and the classroom teacher	Pre: Before and after each experimental break in the classroom. Post: After each experimental break in the classroom.	Accelerometry
Ma et al. (2015) [23]	**IC** “FUNtervals”: eight 20 s periods of VPA (i.e., squats, jumping jacks, scissor kicks, jumping on the spot) separated by 10-s rest periods.	**CC**: 10-min lecture separated from recess by at least 20 min of normal classroom instruction.	10^a^ (4)	3 weeks. On two separate days in random.	After at least 20 min of normal classroom instruction following the recess.	MPA. Aerobic No AC.	Researcher	Pre: In week 1, familiarization. Post: after 10-min researcher-delivered lecture following AB.	Teachers’ control
Niemann et al. (2013) [44]	**IG**: Running on a 400 m track. Participants were not allowed to talk to each other and remained silent.	**CG**: Participants performed sedentary behavior while watching non-arousing scenes. Participants were not allowed to talk to each other and remained silent.	12	NA	After 11:30 am.	VPA. Aerobic. No AC.	NR	Pre: After four normal school lessons just before AB. Post: 5 min after AB.	Control of prior PA in interventions days
Ordóñez et al. (2019) [46]	**IG**: The first two weeks: running a 250-m circuit inside the school; the next four weeks: 500 m; and in the last four weeks: 750 m.	**CG**: No AB.	NA	10 weeks 5 d-w/1 t-d	Between the 2nd and 3rd lesson in the morning.	MPA. Aerobic. No AC.	NR	Pre: At the same time with both groups, just before AB. Post: NR.	Prior familiarization for maintaining MPA
Schmidt et al. (2016) [19]	**IG1** “Combo: high CE + high PA”: PA-based activity of adding numbers. **IG2** “Cognition: high CE + low PA”: A paper -and-pencil trail-making test. **IG3** “Aerobic: low CE + high PA”: Running at different speeds.	**CG** “sedentary + low CE”: Students remained at their desks in the classroom and listened to an age-appropriate story for 10 min to relax and enjoy.	10	NA	After 20 min of German language class (11:15–11:30 a.m.)	MVPA. IG1: Cognitive IG2: no PA. IG3: Aerobic. No AC.	Researchers	Pre: Before AB (10:45–10:55 a.m.). Post: Immediately after AB (11:30–11:40 am).	HR monitors, Borg scale, and self-perceived CE
Schmidt et al. (2019) [24]	**IG1** “Embodied learning condition”: PA-based learning French vocabulary. **IG2** “PA condition”: Movements at the same intensity without academic content.	**CG**: Sedentary teaching style (words were repeated equally as under other conditions).	10	2 weeks 2 d-w/1 t-d	10:00 a.m.–12:00 p.m.	LPA IG1: Cognitive IG2: Aerobic. AC: IG1: earning animals in French; IG2: No.	Trained research student with a video	Pre: Before the beginning of the first learning session. Post: Immediately after the third learning session.	Accelerometry
Tine et al. (2012) [45]	**IG**: Running around an indoor track.	**CG**: Students remained seated and viewed a 12-min film video.	12	NA	2 sessions on separate days during usual gym classes.	VPA. Aerobic. No AC.	Researchers	Pre: Just before AB. Post: One minute after AB.	HR monitors
van den Berg et al. (2016) [51]	**IC1** “Aerobic”: Easy and repetitive movements. **IC2** “Coordination”: Complex movements stressing coordinative skills. **IC3** “Strength”: Dynamic and static body-weight exercises adjusted to the age.	**CC**: 12 min of sitting and listening to an educational lesson about exercise and movement.	12	NA	8:30–10:00 a.m.	LMPA (target: MVPA). IC1: Aerobic. IC2: Coordination. IC3: Strength. No AC.	Researcher, three research assistants, and standardized movie	Pre: Just before AB. Post: Immediately after AB.	HR monitor, familiarization and control of previous bedtime, breakfast, and transport to school
van den Berg et al. (2019) [47]	**IG**: Following three “Just Dance” videos.	**CG**: Nine 10–15 min educational lessons once a week.	10	9 weeks 5 d-w/1 t-d	NR	MVPA. Dancing. No AC.	Teachers	Pre: The week before the intervention started. Post: The following week after the intervention.	AccelerometryTeachers’ control.
Wilson et al. (2016) [49]	**IC** “Active Lesson Breaks” outside the regular classroom, including tag/chasing games, or invasion-type games.	**CC**: Passive lesson break: Participants spent 10 min sitting outside their classroom reading.	10	8 weeks each + 2 weeks of washout. 3 d-w/1 t-d	-	MVPA. Cognitive. AC: based on Take10! and Energizers, or Texas I CAN.	Trained teacher	Pre: 5 min before AB. Post: Immediately after AB.	Accelerometry

AB: active break, AC: academic content; CC: control condition, CE: cognitive engagement, CG: control group, HR: heart rate, IC: intervention condition, IG: intervention group, LMPA: light-to-moderate physical activity, LPA: light physical activity, MPA: moderate physical activity, MVPA: moderate-to-vigorous physical activity, NA: not reported, PA: physical activity; PE: physical education. ^a^ Including time of preparation. * Not AB: Intervention group or condition that did not include ABs. Groups and conditions are reported in bold letters to improve legibility.

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
