# Peer review of "Active School Breaks and Students’ Attention: A Systematic Review with Meta-Analysis"

_brainsci, 2021, doi:10.3390/brainsci11060675_

Round 1

Reviewer 1 Report

Review for manuscript brainsci-1208821-v1 “Active school breaks and students’ attention: A systematic review with meta-analysis” by Á. Infantes-Paniagua et al.

            This manuscript reports a systematic review with meta-analysis of the literature related to the effects of active breaks on students’ attention. This review was based on a search of published interventions involving school-based physical active breaks and objective attentional outcomes given to healthy students of any age. Only interventions that used a counterbalanced cross-over or parallel-groups design and included a control group were considered. Although most results were not significant, the study did find some positive acute and chronic effects of active breaks on attention as indexed via task accuracy, concentration, inhibition, and the ability to orient and sustain selective attention. The authors found no evidence that attention was impaired by active breaks. The authors argue that the null results are primarily due to the small number of studies included in the analysis and the heterogeneity in design across the analyzed studies.

I found this study to be an interesting review and meta-analysis that address a timely and important real-world topic. The present study is timely given that no prior meta-analysis has been performed on the influence of attentional breaks in attention-related mental information processing. The topic of the present study is important because it may demonstrate systematic evidence of an easily-implementable method to improve children’s mental information processing and their ability to learn information in an educational setting. In my assessment, the review and meta-analysis methodology is satisfactory and has been reported adequately. That said, I do have a few questions and/or clarification requests about the present manuscript that, if addressed, could make the study’s overall readability and impact much stronger.

  1. 3, lines 128 – 129 and Table 1: The authors state that “according to the P.I.C.O.S. approach, the inclusion and exclusion criteria for this systematic review and meta-analysis can be found in table 1”. What does the “P.I.C.O.S” acronym stand for? I cannot find this anywhere in the manuscript. Please explicitly define all major and/or novel acronyms used in the paper. Also, why was this approach utilized in this study?

p.4, lines 158 – 160: The authors state “The methodological quality of studies was assessed using the RoB 2 for RCTs [31], 158 the tool’s supplements for cluster randomized trials (CRTs) [32] and for crossover trials 159 [33]”. What does the “ROB 2” acronym stand for? Please explicitly define all major and/or novel acronyms used in the paper.

p.4, lines 165-166; Could the authors please add a statement regarding the relevance and or importance of “an ‘intention-to-treat’ effect approach for the assessment of methodological quality?

p.5, lines 210 – 212: The authors state “Overall, 210 ABs consisted in aerobic or coordinative moderate PA (MPA) [6,23,35,41,46,51], vigorous 211 PA (VPA) [6,44,45], or MVPA [19,41–43,47,50]”. Although the “MVPA” acronym is defined later in the manuscript, it should be defined at first use. Please explicitly define all major and/or novel acronyms used in the paper.

p.6, Figure 1: What does the “PRISMA” acronym stand for in the figure caption? Please explicitly define all major and/or novel acronyms used in the paper.

p.24, lines 408 – 409; The authors state that “In light of the above discussion, another key factor that might moderate the effects of ABs is the person who applies the AB”. In the meta-analysis, is it possible to account or weight for whether an individual study had a teacher with an active role in the AB interventions? This would provide empirical support for this suggestion.

Reviewer 2 Report

The potential benefit of physical activity is a current and important topic. This systematic review and meta-analysis were designed to evaluate the evidence for improved academic engagement by 7 to 13 year old students after short physical activity breaks during class time.  

Abstract

Line 40, The phrase “the responsibility of the active break..” does not convey the meaning given in the main text.  Perhaps use a phrase like this: the leader of the active break.     

Line 121, Inclusion criteria Please state how the different stages of development of the young people were taken into account in the analysis of the effect of the intervention.

Lines 96, 97.  Breaks of more than 20 minutes would start to affect the learning time in a lesson.  This should be discussed.

Line 171, Consider changing the name of this sub-heading which is currently a confusing mixture.

Line 200.  Justify grouping 7 to 13 year olds together when they have such different needs due to different stages of development.

Line 220, The quality of figure 1 is poor, including the carriage-return markers showing in each box, which must be removed.

Table 2 and Table 3. Change the structure of the tables so that the results column is easier to read.   Consider also changing the orientation of the table layout to landscape.

Table 4. Restructure to enable the column headed type of AB can be read.

Line 255, Give the synthesis in text here, do not direct the reader to go back to the table of results which simply presents the results from each study. The synthesis is the interpretation of the results as a whole and belong in the text, not the table.

Discussion

Lines 361 – 363 The meta-analysis showed no statistical difference between AB and control for inhibition (Fig 2), concentration (Fig 3) and shifting (Fig 5).  The trends for inhibition and for concentration were towards AB, and the trend for shifting was towards the control. However, all three meta-analyses showed the confidence interval included zero, so no conclusions from these results should be drawn. There was significant difference between AB and control for selective attention (Fig 4) and the confidence interval did not include zero, so this set of results should be discussed.  Please amend the second sentence of the discussion to reflect this information accurately.   

Line 383, A large part of the meta-analysis presented here does not unfortunately provide supportive evidence for anything.  One aspect here, the selective attention, support the conclusions on cognitive function by Donnelly et al 2016. Please amend the text to be specific.

Please use a native English speaker or a professional editing service.  Some examples are listed below, but there were too many to list all of them. 

Line 101, correct word order.  “…duration of the cognitive benefits that remain after the bout of PA, …”

Line 104, word missing. “Finally, the person responsible for delivering the AB might also play a key role,…”

Line 114, To the best of our knowledge, there is only one previous meta-analysis that examined this [2], and it only focused on overall cognitive- or academic-related outcomes in 6-9 year-old students.

Line 123, correction required. “..followed established international guidelines.” 

Table 1,  Increase the space between the columns of information. Set the justification the left-hand justification in each column to make it easier to read.

Line 150 Correct the verb. “The following categories of information were extracted from included articles.”

Line 154 Incorrect English.  “…responsible of AB and protocol)…”  Perhaps it is meant to say “..who is responsible for the AB and protocol..”   However, it is unclear, so please correct the phrase to state what you mean.

Line 159, Correct the grammar and word order of the phrase: “…the tool’s supplements for cluster randomised trials ……….was….”  The verb should be were to match the plural noun supplements.  

Line 161, Correct the text.  “…21 items that enable the assessment of the risk of bias….” 

At this point it was evident a full review of the English is required, so no further detailed comments are made. Please have all the text checked.

Round 2

Reviewer 1 Report

The authors have satisfactorily addressed all of my concerns.

Author Response

On behalf of the authors of the manuscript entitled “Active school breaks and students’ attention: A systematic review with meta-analysis” (Manuscript ID: brainsci-1208821), I would like to let you know that we are really pleased for the positive feedback received from the reviewer. The previous reviews have let us improve the comprehension and readability of our work; we are really grateful for that.

Thank you both.

Best regards,

Álvaro Infantes-Paniagua

Reviewer 2 Report

Thank you for addressing each of the points raised and for using an excellent service for the English language.  The result is a valuable systematic review and meta analysis for educators and researchers involved in this very important area.  The final very small changes I have suggested on the attached pdf are simply to help those who are unfamiliar with the interpretation of forest plots to understand the significance of each of your excellent diagrams.

Author Response

On behalf of the authors of the manuscript entitled “Active school breaks and students’ attention: A systematic review with meta-analysis” (Manuscript ID: brainsci-1208821), I would like to let you know that we are really pleased for the positive feedback and detailed comments received from the reviewer. These have let us improve the comprehension and readability of our work; we are really grateful for that. Accordingly, we have made all the suggested changes and tracked them on the following table (please see the attached document). We have accepted all the previous changes on the track changes option except for the ones addressed in this second round of reviews. All the references to the lines of the manuscript were made considering that the track changes option is activated. We hope that the improvements made in the manuscript meet the quality of the journal and the expectations of the reviewer.

Thank you both.

Best regards,

Álvaro Infantes-Paniagua
